# Diagnosis of *Clostridioides difficile* infection by analysis of volatile organic compounds in breath, plasma, and stool: A cross-sectional proof-of-concept study

Teny M. John[1,2]*, Nabin K. Shrestha[1], Gary W. Procop[3], David Grove[4], Sixto M. Leal, Jr[3,5], Ceena N. Jacob[6,7], Robert Butler[8], Raed Dweik[4]

1 Department of Infectious Diseases, Cleveland Clinic, Cleveland, Ohio, United States of America,
2 Department of Infectious Disease, The University of Texas MD Anderson Cancer Center, Houston, Texas, United States of America, 3 Department of Laboratory Medicine and Pathology, Cleveland Clinic, Cleveland, Ohio, United States of America, 4 Department of Pulmonary Medicine and Critical Care, Respiratory Institute, Cleveland Clinic Foundation, Cleveland, Ohio, United States of America, 5 Department of Laboratory Medicine and Pathology, University of Alabama at Birmingham, Birmingham, Alabama, United States of America, 6 Department of Internal Medicine, Cleveland Clinic Foundation, Cleveland, Ohio, United States of America, 7 Department of Internal Medicine, The University of Texas Health Science Center at Houston, Houston, Texas, United States of America, 8 Quantitative Health Sciences, Cleveland Clinic Foundation, Cleveland, Ohio, United States of America

* tmjohn1@mdanderson.org

**Data Availability Statement:** The analysis dataset and code to reproduce the results are available in the publicly available site https://osf.io/5hv49/.

## Abstract

*Clostridioides difficile* infection (CDI) is an important infectious cause of antibiotic-associated diarrhea, with significant morbidity and mortality. Current diagnostic algorithms are based on identifying toxin by enzyme immunoassay (EIA) and toxin gene by real-time polymerase chain reaction (PCR) in patients with diarrhea. EIA's sensitivity is poor, and PCR, although highly sensitive and specific, cannot differentiate infection from colonization. An ideal test that incorporates microbial factors, host factors, and host-microbe interaction might characterize true infection, and assess prognosis and recurrence. The study of volatile organic compounds (VOCs) has the potential to be an ideal diagnostic test. The presence of VOCs accounts for the characteristic odor of stool in CDI but their presence in breath and plasma has not been studied yet. A cross-sectional proof-of-concept study analyzing VOCs using selected ion flow tube mass spectrometry (SIFT-MS) was done on breath, stool, and plasma of patients with clinical features and positive PCR for CDI (cases) and compared with patients with clinical features but a negative PCR (control). Our results showed that VOC patterns in breath, stool, and plasma, had good accuracy [area under the receiver operating characteristic curve (ROC) 93%, 86%, and 91%, respectively] for identifying patients with CDI.

## Introduction

*Clostridioides* (formerly known as *Clostridium) difficile* infection (CDI) is an important infectious cause of antibiotic-associated diarrhea [1]. The annual burden of infections in the United

**Funding:** This research was supported by Cleveland Clinic Lerner Research Institute Research Program Committee grant (grant number: 290, awarded to Teny M. John, March 13, 2018) The funders had no role in study design, data collection, and analysis, decision to publish, or preparation of the manuscript.

**Competing interests:** The authors have declared that no competing interests exist.

States was close to 16,000 in 2018, with in-hospital mortality of 8.4% in patients older than 65 [2]. CDI is predominantly a healthcare-associated infection with prior antibiotic exposure considered a significant risk factor [1]. The clinical spectrum ranges from asymptomatic colonization, in 3%-26% of adult inpatients, to severe, life-threatening, and fulminant colitis. The current diagnostic strategy that involves a multi-step test using toxin and PCR in the stool has many challenges [3]. Although highly sensitive and specific, PCR tests alone cannot differentiate infection from colonization, a common scenario in clinical practice. EIA test for stool toxin lacks sensitivity, with values as low as 45% being reported [4]. Turn-around time from ordering a test to result may take up to 24 hours as stool samples cannot be produced 'on-demand,' and samples need to be transported to the lab before being analyzed [5]. Rapid, sensitive, specific, point-of-care testing methods are required for early diagnosis of CDI.

Volatile organic compounds (VOCs) are aromatic hydrocarbon end product metabolites of physiological and pathophysiological processes [6]. VOCs are transported through the blood from different organs to the lungs and subsequently exhaled. Patients with CDI have been noted to have a characteristic odor ('horse barn odor') from the assemblage of VOCs present [5, 7, 8]. Previous studies using gas chromatography-mass spectrometry (GC-MS) characterized this 'volatile molecular signature' in the stool of patients with CDI [7]. Rees *et al.* identified 77 molecules using headspace VOC analysis from *C.difficile* cultures. Another recent study of stored stool samples from 53 cases and 53 controls, using thermal desorption-gas chromatography-time-of-flight gas chromatography, identified seven compounds (propan-1-ol, 3-methylbutanal, ethyl propionate, hexanoic acid, 4-methyl phenol, dodecane, and indole) indicative of CDI with a ROC >0.7 [5]. Selected ion flow tube mass spectroscopy (SIFT-MS) enables the faster measurement of lower concentrations (parts per billion or even trillion) of VOCs in clinical samples [9]. SIFT-MS technology has shown high discriminatory capacity in other syndromes like inflammatory bowel disease, nonalcoholic fatty liver disease, and pulmonary artery hypertension, but has not been used to study CDI [10–12]. The purpose of this study was to determine if VOCs in stool, blood, and breath, of patients with CDI, as measured by SIFT-MS, differ from those in age and gender-matched controls without CDI.

## Materials and methods

This cross-sectional study enrolled patients > 18 years old with diarrhea who had stools tested for *Clostridioides difficile* by PCR [done using BD GeneOhm™ (BD Diagnostics, Franklin Lakes, NJ)]. Written informed consent was obtained from all patients before enrollment. Patients with >3 episodes of diarrhea in the preceding 24 hours and an illness suggestive of *C. difficile* infection (abdominal pain, fever, elevated WBC count) with a stool specimen positive for *C. difficile* by PCR were considered to have CDI. The single best age and gender-matched patient, with liquid stools but negative *C. difficile* PCR on the same day, was selected as a control for each patient included in this study. Consecutive cases and controls identified during working days (Monday through Friday) were included. Those without a clinical illness compatible with CDI, those who refused or were unable to give informed consent (e.g., due to intubation, encephalopathy, delirium, or pharmacologic sedation), those requiring supplemental oxygen, and those with CDI in the previous four weeks were excluded. The study was approved by the Cleveland Clinic Institutional Review Board, IRB # 18–030.

Stool samples sent to the microbiology laboratory for *C.difficile* PCR testing, and plasma samples drawn within 24 hours of stool collection were identified, and residual stool specimen and 100 μL of residual plasma specimen were obtained. Breath samples were collected at the patient's bedside within 24 hours of collection of the stool specimen. Initially, tidal volume exhalation was done to clear residual air from the anatomic dead space, followed by a deep

breath through a disposable micro-filtered mouthpiece, which prevented exposure to viral and bacterial pathogens in ambient air and eliminated exogenous VOCs, followed by tidal volume exhalation back through the mouthpiece. Exhaled breath was collected in a Mylar balloon bag. All samples (breath, stool, and plasma) were incubated at 37°C for 30 minutes before analysis to desorb the VOC's from the surface of the container. For stool and plasma samples, 20 mL of headspace gas was removed from the vials using a glass syringe. For breath samples, the Mylar bag was connected to the mass spectrometry device directly.

The gas from samples was analyzed using a VOICE200 SIFT-MS instrument *(Syft Technologies Ltd, Christchurch, New Zealand)*. VOCs were measured in real-time after they underwent chemical ionization using $H_3O^+$, $NO^+$ and $O_2^+$ precursor ions [13]. Product ion masses of VOC analytes were detected and counted by a downstream mass spectrometer (MS). A complete mass spectrum was obtained for mass-to-charge ratio values between 14–200 to identify significant peaks of product ion masses representing VOCs relating to CDI. The count rate of product ions was directly proportional to the concentration of the VOC.

The laboratory personnel analyzing the samples, and the microbiology technologists aliquoting the stool and plasma samples, were blinded to the stool *C. difficile* test results. The distribution of clinical and the VOC analyte variables were compared for CDI and control patients. Prediction models were developed for each sample type, based on K-nearest neighbors (KNN) regression, to classify each observation into CDI or not CDI, using VOC analytes only, and separately using clinical and VOC analyte variables. Model performances were validated using 5-fold cross-validation. Receiver operating characteristics (ROC) curves were generated for the prediction for each sample type. Analyses were done using R version 4.0.5.

## Results

Of 67 patients with positive stool *C.difficile* PCR screened for inclusion in the study, 36 were excluded for various reasons (Fig 1). Each of the remaining 31 patients had a matched control.

### Baseline characteristics

The CDI and non-CDI groups were comparable for the examined clinical variables, except that a higher proportion of the cases had heart failure (Table 1).

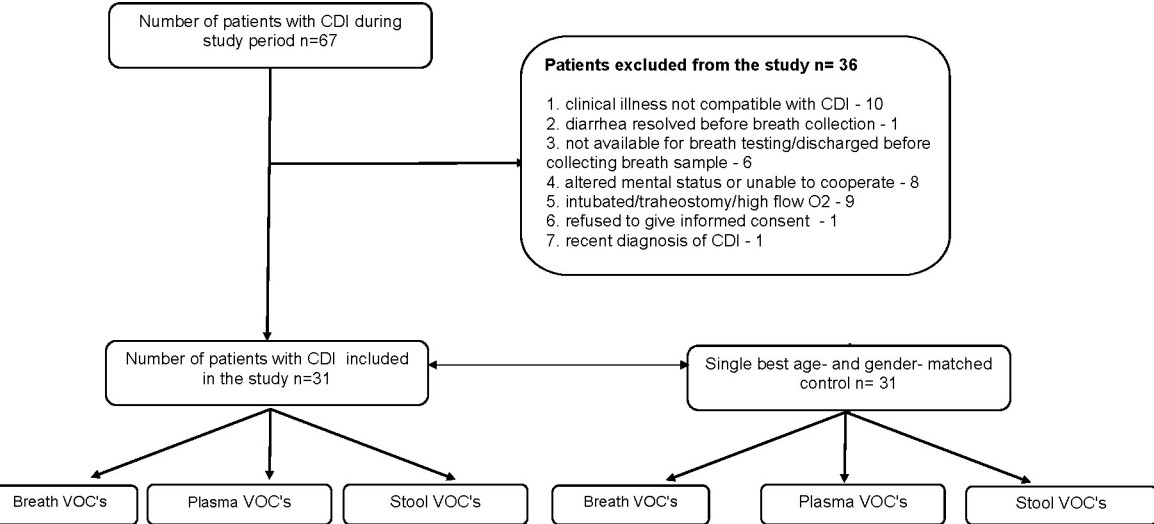

**Fig 1. Study flow chart.** Flow chart showing the number of patients screened, patients included, and their comparison to age- and gender-matched controls.

**Table 1. Baseline demographic characteristics.**

| Characteristic | Cases (n = 31) | Controls (n = 31) | *P* value |
|---|---|---|---|
| Age (years) | 56.9 ± 15.1 | 52.8 ± 15.3 | 0.29 |
| Race | | | 0.71 |
| Caucasian | 26 (84%) | 27 (87%) | |
| African American | 5 (16%) | 3 (10%) | |
| Others | 0 (0%) | 1 (3%) | |
| Male | 15 (48%) | 15 (48%) | 0.99 |
| Body Mass Index (kg/m$^2$) | 28.8 ± 9.6 | 29.0 ± 7.2 | 0.94 |
| Comorbidities | | | |
| Diabetes Mellitus | 9 (29%) | 7 (23%) | 0.56 |
| Coronary Artery Disease | 6 (19%) | 2 (6%) | 0.26 |
| Heart Failure | 6 (19%) | 0 (0%) | 0.02 |
| Chronic Obstructive Pulmonary Disease | 2 (6%) | 1 (3%) | 0.99 |
| Chronic Kidney Disease | 4 (13%) | 5 (16%) | 0.99 |
| Chronic Liver Disease | 1 (3%) | 3 (10%) | 0.61 |
| Inflammatory Bowel Disease | 2 (6%) | 7 (23%) | 0.15 |
| Malignancy | 12 (39%) | 5 (16%) | 0.05 |
| History of Transplant | 5 (16%) | 9 (29%) | 0.22 |
| Solid Organ Transplant | 4 | 5 | |
| Hematopoietic Stem Cell Transplant | 1 | 3 | |
| Concurrent Infection(s) | 11 (35%) | 9 (29%) | 0.59 |
| Smoking | | | 0.99 |
| Current Smoker | 5 (16%) | 5 (16%) | |
| Ex-Smoker | 9 (29%) | 9 (29%) | |
| Non-Smoker | 17 (55%) | 17 (55%) | |
| Alcoholism | 10 (32%) | 10 (32%) | 0.99 |
| *Clostridioides difficile* Severity* | | | |
| Non–severe | 18 (58%) | NA | — |
| Severe | 8 (26%) | NA | — |
| Fulminant | 5 (16%) | NA | s— |
| Prior history of *Clostridioides difficile* (> 4 weeks ago) | 4 (13%) | 1 (3%) | 0.35 |
| Peak WBC Count (x 10$^9$/ L) | 11.5 (5.2, 20.6) | 10.1 (5.3, 12.9) | 0.35 |
| Peak Serum Creatinine (mg/dL) | 1.1 (0.8, 4.0) | 0.9 (0.8, 1.4) | 0.10 |
| Nadir Serum Albumin (g/dL) | 2.9 (2.4, 3.6) | 2.8 (2.3, 3.3) | 0.49 |

Descriptive statistics reported as either mean ± standard deviation, median ($Q_1$, $Q_3$) or count (%)

Means are compared with t-tests, medians compared with Wilcoxon rank-sum tests and proportions compared with chi-square or Fisher's exact test as appropriate.

Abbreviations: NA, not applicable. Non-severe disease is defined as leukocytosis with a white blood cell count of ≤15 000 cells/mL and a serum creatinine level <1.5 mg/dL, severe disease—leukocytosis with a white blood cell count of ≥15 000 cells/mL or a serum creatinine level >1.5 mg/dL and fulminant disease–presence of hypotension or shock, ileus, megacolon.

## Volatile organic compounds

Product ion concentrations in patients with and without CDI are compared in S1–S3 Tables, for breath, stool, and plasma samples, respectively. Those product ions that are statistically significant (p < 0.05 on univariable analysis) with good branching ratios (50% or greater branching ratio), that are identified by multiple reagent ions and that had a possible identification (putative ID) are depicted in Table 2. An example of mass spectra using H3O+ precursor from a patient with CDI is provided in S1 Fig.

**Table 2. Product ions that were significantly different in breath, stool and plasma.**

| Product ions in breath | Putative ID | Product ions in stool | Putative ID | Product ions in plasma | Putative ID |
|---|---|---|---|---|---|
| H3O+65+ | ethanol | H3O+185+ | 1,2,4-trichlorobenzene, 1,1,2,2-tetrachloroethane | O2+180+ | 1,2,4-trichlorobenzene |
| H3O+70+ | putresccine | H3O+187+ | 1,2,4-trichlorobenzene, 1,1,2,2-tetrachloroethane | O2+182+ | 1,2,4-trichlorobenzene |
| NO+45+ | ethanol | | | O2+184+ | 1,2,4-trichlorobenzene |
| NO+63+ | ethanol | | | | |
| NO+87+ | putrescine | | | | |
| NO+174+ | dimethyl fumarate | | | | |
| O2+29+ | formaldehyde | | | | |
| O2+45+ | ethanol | | | | |
| O2+63+ | ethanol (water cluster) | | | | |
| O2+65+ | ethanol | | | | |
| O2+113+ | dimethyl fumarate | | | | |

(P- value < 0.05) with their possible identification. (Putative IDs are for those product ions having 50% or greater branching ratio and the compounds that are significantly identified by multiple reagent ions with good branching ratios.)

### Distribution of VOC concentrations

There were differences in concentrations of various product ions in CDI cases and controls. A heatmap with cluster dendrograms showing the distribution of concentrations of the various product ions in breath samples across the study participants is shown in Fig 2. This shows some clustering according to whether or not patients had CDI.

### Diagnosis of CDI using VOC analysis

For predicting the presence or absence of CDI, the optimal KNN classifier model was achieved with k = 7, 9, and 9, for breath, stool, and plasma samples, respectively. The areas under the curve (AUCs) for predicting the presence of CDI from these models were 93%, 86%, and 91%, for breath, stool, and plasma samples, respectively (Fig 3). Accuracy was better if clinical variables were considered in addition to product ion concentrations in the model (AUC of 95%, 85% and 94% for breath, stool and plasma samples respectively) (S2 Fig). Receiver operating characteristic (ROC) curves for identification of CDI using breath, stool, and plasma samples, are shown in Fig 3.

Model accuracy was not appreciably better if only positives with *C. difficile* PCR cycle threshold (CT) < 30 cycles were included (S3 Fig).

### Discussion

Prior studies using SIFT-MS technology have demonstrated the utility of breath analysis in a wide range of non-infectious conditions, including inflammatory bowel disease, non- alcoholic fatty liver disease, and fibrosis associated with chronic liver disease [10]. This proof-of-concept study showed that VOC patterns in breath, stool, and plasma, are different in patients with and without CDI, and that analysis of these differences has good accuracy in making a diagnosis of CDI. One molecule in breath that was significantly identified by multiple reagent ions that deserves special mention is putrescine. Putrescine belongs to the family of polyamines that is involved in arginine and ornithine metabolism [14]. They are present inside the

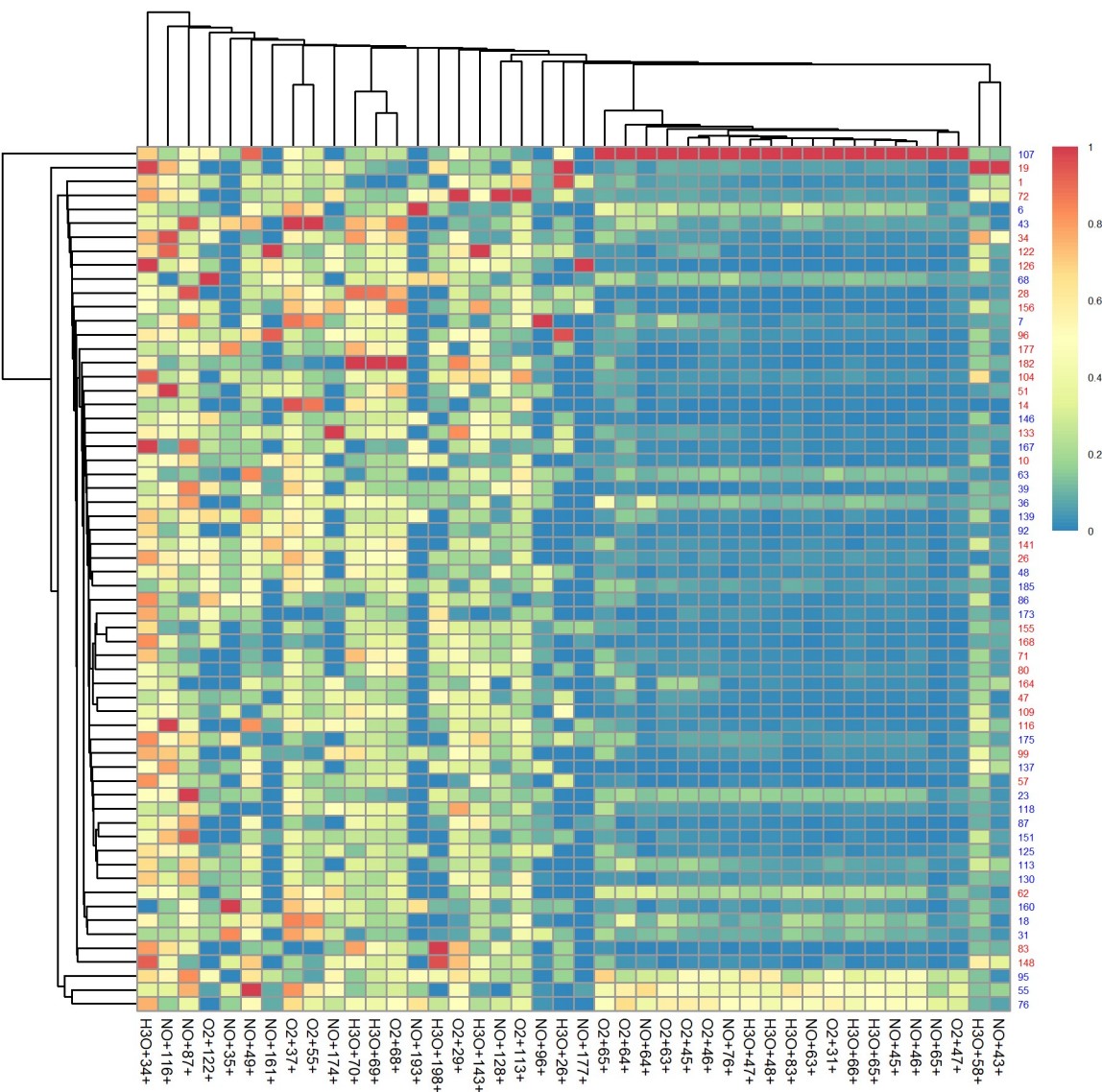

**Fig 2. Relative abundance of breath VOCs in CDI.** This heatmap with cluster dendrogram shows the distribution of concentration of various product ions in patients with (red labels) and without CDI (blue labels). Scaled relative concentrations of product ions are depicted in the heatmap by colors on a spectral scale. The linkage method used for hierarchical clustering was the nearest neighbor method.

cells of all mammalian species, where they are involved in various cellular functions including protein synthesis and cell growth [15]. Arginine increases the growth of *Clostridium difficile* in vitro, and seems to have a role in toxin production in some strains [16]. Larger studies are required to validate this finding and it is also unclear why these molecules are not observed in stool or plasma.

The small sample size is an important limitation of our study. Larger studies to validate our findings are needed. It is possible that molecules outside the range of those analyzed by SIFT-MS may provide further discrimination. But this proof-of-concept study shows that there is promise in analyzing breath to diagnose CDI. Another limitation is that breath collection using Mylar bags and transportation to the laboratory may have resulted in the loss of

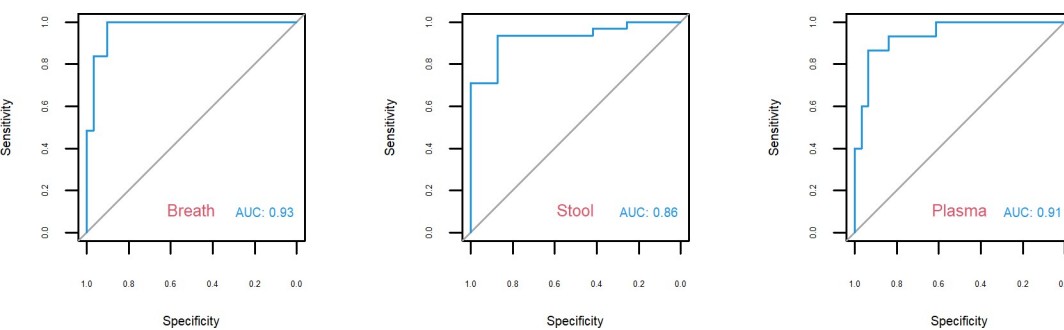

**Fig 3. Receiver operating characteristic (ROC) curves with area under the curve for breath, stool and plasma samples respectively.**

VOCs. More robust breath collection devices may mitigate this limitation. Lastly, the small number of patients with chronic conditions made it difficult to interpret the confounding effect of comorbid conditions.

In conclusion, this proof-of-concept study shows that VOC patterns in breath, stool, or plasma, using SIFT-MS had good accuracy for identifying patients with CDI. If validated in future studies, easier to collect and readily available breath samples can be used for rapid, bedside diagnosis of CDI.

## Supporting information

**S1 Fig. An example of mass spectra using H3O+ precursor from a patient with CDI.** (TIF)

**S2 Fig. Model accuracy for breath, stool and plasma, if clinical variables are considered in addition to VOC analyte variables.** (TIF)

**S3 Fig. Model accuracy for breath, stool, and plasma specimens, if only positives with *C. difficile* PCR cycle threshold (CT) < 30 cycles are included.** (TIF)

**S1 Table. Comparison of product ion concentrations in breath of patients with and without CDI.** (CSV)

**S2 Table. Comparison of product ion concentrations in stool of patients with and without CDI.** (CSV)

**S3 Table. Comparison of concentrations in plasma of patients with and without CDI.** (CSV)

## Acknowledgments

We thank Ms. Laura Doyle and Deborah Wilson for laboratory support in aliquoting the stool and plasma samples, Mr. Mike Sutton for providing daily *C. difficle* reports. We want to acknowledge Dr. Leslie Silva (Syft Technologies) for providing putative IDs.

## Author Contributions

**Conceptualization:** Teny M. John, Nabin K. Shrestha, Gary W. Procop, David Grove, Sixto M. Leal, Jr, Ceena N. Jacob, Raed Dweik.

**Data curation:** Teny M. John, Nabin K. Shrestha, David Grove, Sixto M. Leal, Jr.

**Formal analysis:** Nabin K. Shrestha, David Grove, Sixto M. Leal, Jr, Robert Butler, Raed Dweik.

**Funding acquisition:** Teny M. John.

**Investigation:** Teny M. John, Nabin K. Shrestha, Gary W. Procop, David Grove, Raed Dweik.

**Methodology:** Teny M. John, Nabin K. Shrestha, Gary W. Procop, David Grove, Sixto M. Leal, Jr, Robert Butler, Raed Dweik.

**Project administration:** Teny M. John, Nabin K. Shrestha, Gary W. Procop, Raed Dweik.

**Resources:** Teny M. John, Nabin K. Shrestha, Gary W. Procop, David Grove, Sixto M. Leal, Jr, Robert Butler, Raed Dweik.

**Software:** Nabin K. Shrestha, Raed Dweik.

**Supervision:** Teny M. John, Nabin K. Shrestha, Gary W. Procop, Raed Dweik.

**Validation:** Teny M. John, Nabin K. Shrestha, Gary W. Procop, David Grove, Raed Dweik.

**Visualization:** Teny M. John, Nabin K. Shrestha, Gary W. Procop, David Grove, Raed Dweik.

**Writing – original draft:** Teny M. John, Nabin K. Shrestha, David Grove, Sixto M. Leal, Jr, Ceena N. Jacob, Raed Dweik.

**Writing – review & editing:** Teny M. John, Nabin K. Shrestha, Gary W. Procop, David Grove, Sixto M. Leal, Jr, Ceena N. Jacob, Robert Butler, Raed Dweik.

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
