## [Decision Letter · Decision Letter 0]

4 Nov 2020

PONE-D-20-23689

Diagnosis of Clostridioides difficile infection by analysis of volatile organic compounds in breath, plasma, and stool – a cross-sectional proof-of-concept study.

PLOS ONE

Dear Dr. John,

Thank you for submitting your manuscript to PLOS ONE. After careful consideration, we feel that it has merit but does not fully meet PLOS ONE’s publication criteria as it currently stands. Therefore, we invite you to submit a revised version of the manuscript that addresses the points raised during the review process.

We had comments from 3 reviewers who all believed the work is exciting.  They each have provided comments and I request that you prepare a response addressing each of the comments.

We look forward to receiving your revised manuscript.

Kind regards,

Timothy J Garrett, PhD

Academic Editor

PLOS ONE

Additional Editor Comments:

Thank you for your submission. I have carefully reviewed all the comments from the reviewers and agree that a major revision is needed. Please carefully consider and appropriately respond to each comment.

Journal Requirements:

2. Thank you for including the following ethics statement on the submission details page:

'Approved by Cleveland Clinic institutional review board, IRB# 18-030.

Written informed consent obtained from all participants'

Please also include this information in the ethics statement in the Methods section of your manuscript.

3.  We noted in your submission details that a portion of your manuscript may have been presented or published elsewhere.

"Findings were presented at ID week 2018 as a poster and abstract was published in Open Forum of Infectious Diseases."

Please clarify whether this publication was peer-reviewed and formally published. If this work was previously peer-reviewed and published, in the cover letter please provide the reason that this work does not constitute dual publication and should be included in the current manuscript.

Reviewers' comments:

Reviewer's Responses to Questions

**Comments to the Author**

1. Is the manuscript technically sound, and do the data support the conclusions?

Reviewer #1: No

Reviewer #2: Partly

Reviewer #3: Yes

2. Has the statistical analysis been performed appropriately and rigorously? 

Reviewer #1: I Don't Know

Reviewer #2: Yes

Reviewer #3: Yes

3. Have the authors made all data underlying the findings in their manuscript fully available?

Reviewer #1: Yes

Reviewer #2: No

Reviewer #3: Yes

4. Is the manuscript presented in an intelligible fashion and written in standard English?

Reviewer #1: Yes

Reviewer #2: Yes

Reviewer #3: Yes

5. Review Comments to the Author

Reviewer #1: Notable point on the article is the use of paired controls to reduce if not eliminate any bias due to other factors unrelated to the disease. Some typographical errors found but not in a way that can influence the article.

Line 125 Capitalize C on Celsius

Fig 1 “Stautus” to “status”

However, a major issue that is detrimental to the study is the use of breath metabolites as metrics to differentiate or classify stool and plasma samples. It is not surprising that there is low accuracy in these samples, and therefore invalid comparison. The authors have even stated that there is a characteristic odor of infected stool, which also suggests to the readers that there may be challenges in using SIFT MS for detection of these characteristic compounds. A helpful suggestion is to reclassify the stool and plasma samples using their intrinsic metabolites.

Reviewer #2: The authors provide a very brief manuscript describing the use of SIFT-MS to analyze molecules released from different samples from patients and controls for the detection of C. difficile. The paper has some interesting data, but needs to be expanded on a number of different points to enable clearer understanding of the need for the research, reproducibility of the methods for data analysis, more thorough evaluation of the results, and confirmation that the results support the conclusions that are presented in the manuscript. Major critiques include:

1. Please provide the individual data for the patients including clinical status and values for each analyte observed in SIFT-MS.

2. Example mass spectra for each sample type would be helpful to the reader.

3. Representation of the data using principal component analysis, hierarchical clustering with a heat map, or some similar approach would enable the authors’ claims of separation of groups (C. diff and control) to be evaluated more effectively. At the current time, the claims for separation of the groups can’t be evaluated by the way the data are presented.

The draft is well written, but minor editing for spelling and grammar is needed.

Additional specific requests for revisions are included below.

Abstract:

Please add detail about why C. difficile is important and what methods are currently available for its detection to describe the need for the research presented here.

Introduction:

In the previous GC-MS work on C. difficile, which compounds could be used to differentiate infected people from controls? Are they detected with SIFT-MS?

Table 1: Do the authors expect that the confounding variables from other conditions (e.g. cancer) will impact the differences between the groups? Data may need to be presented in the figures with color codes to indicate patients these other chronic diseases.

Why was the K nearest neighbors strategy selected? Please explain the rationale.

Do any of the molecules detected in SIFT-MS have potential as individual biomarkers? This point is addressed in part by the comparisons and p values in the supplemental tables, but it should also be presented in the results section of the manuscript.

Supplemental Figures and Tables need captions on the page with the data or image.

Supplemental Tables need headings to explain the data, which are presented as average and standard deviation but not labeled.

Reviewer #3: Here the authors present a clinical study which shows that breath samples could be used to diagnose if someone has a CDI. In addition to breath, they also analyzed the headspace of both plasma and stool. Both of the other diagnostic mediums had poor predictive power which could have been cause by the loss in VOCs during the sample handling process. I find the scientific work interesting and generally well done. With that said, this paper would benefit if the authors would expand the introduction and addressed concerns below.

Introduction: This section is very short. It almost reads like an abstract. The authors should do a better job introducing breath analysis, its prior use for CDI, and the various analytical techniques. The section could end on the rationale used for the selection of SIFT-MS. This section should be at least two paragraphs longer.

Materials and Methods:

Why were weekend samples excluded? Please do a better job explaining the rationale.

Why were samples incubated before analysis? Is this common practice?

Table 1: Does not significantly contribute to the paper. I would recommend moving this to the supplemental section or shortening it for the main text.

Supplemental Tables 1-3. Given that these numbers are at the heart of the study, I would move supplemental table 1 into the main text. For all supplemental tables, what are the units? What is pos and neg? CDI pos, CDI neg? Please label more clearly.

Supplemental Figure 1 and 2 need figure legends.

Results:

I understand that the entire 22 VOC panel was used for the ROC curves. However, was there a subset of VOCs in the panel that was MORE predictive than the full panel for each type of sample? If so what?

Other general comments:

Why was blood plasma used? Why not whole blood? The sample processing of whole blood to plasma would significantly change and or remove important VOCs.

6. PLOS authors have the option to publish the peer review history of their article (what does this mean?). If published, this will include your full peer review and any attached files.

Reviewer #1: No

Reviewer #2: No

Reviewer #3: No

---

## [Author Response · Author response to Decision Letter 0]

2 Apr 2021

Thank you for reviewing our manuscript and providing valuable suggestions. We hereby respond to reviewer and editors comments in a question and response format.

1. 'Approved by Cleveland Clinic institutional review board, IRB# 18-030. Written informed consent obtained from all participants. Please also include this information in the ethics statement in the Methods section 

Response from authors: Thank you for your suggestion. We included these key sentences in the methods section.

2. We noted in your submission details that a portion of your manuscript may have been presented or published elsewhere. "Findings were presented at ID week 2018 as a poster and abstract was published in Open Forum of Infectious Diseases." Please clarify whether this publication was peer-reviewed and formally published. If this work was previously peer reviewed and published, in the cover letter please provide the reason that this work does not constitute dual publication and should be included in the current manuscript. Response from authors: Findings were presented as a poster at ID week 2018, the abstract of it was published in a supplement of OFID as a conference abstract. While conference abstracts are reviewed by the scientific committee, they are not considered peer-reviewed.

Reviewer # 1 comments:

1. Some typographical errors found but not in a way that can influence the article. Line 125 Capitalize C on Celsius and Fig 1 “Stautus” to “status” 

Response from authors: Thank you; both these typographical errors are corrected. 

2. However, a major issue that is detrimental to the study is the use of breath metabolites as metrics to differentiate or classify stool and plasma samples. It is not surprising that there is low accuracy in these samples, and therefore invalid comparison. The authors have even stated that there is a characteristic odor of infected stool, which also suggests to the readers that there may be challenges in using SIFT MS for detection of these characteristic compounds. A helpful suggestion is to reclassify the stool and plasma samples using their intrinsic metabolites. 

Response from authors: Thank you for your suggestion. The characteristic odor of stool in patients with Clostridioides difficile infection (CDI) is well known, and it is thought to be secondary to metabolites in the stool. Studies have shown that dogs can scent stool odor and correctly classify CDI. Multiple invitro studies based on stool cultures also confirm the presence of VOCs in the stool. We want to clarify this study has not focused on classifying stool and plasma samples based on breath metabolites. On the other hand, we have studied cross-sectionally collected stool, plasma, and breath samples (all collected within a 24-hour time frame) of patients with CDI. We think the lack of signal in stool and plasma is due to using left-over samples that was not stored appropriately resulting in loss of VOCs.

Reviewer #2:

1. Please provide the individual data for the patients including clinical status and values for each analyte observed in SIFT-MS.

Response from authors: Thank you for your suggestion. Clinical data is provided in Table 1. This has already been deposited in a public repository at https://osf.io/5hv49/. The repository contains the analysis dataset and the analysis to reproduce the results 

2. Example mass spectra for each sample type would be helpful to the reader. Response from authors: This is a very good suggestion. A sample mass spectra for each type of sample – namely breath, plasma and stool for a patient is added as a supplementary figure. 

3. Representation of the data using principal component analysis, hierarchical clustering with a heat map, or some similar approach would enable the authors’ claims of separation of groups (C. diff and control) to be evaluated more effectively. At the current time, the claims for separation of the groups can’t be evaluated by the way the data are presented.

Response from authors: Thank you for this suggestion. We have provided a cluster dendrogram that shows clustering of patients with CDI(in red). (Figure 3)

4. Abstract :Please add detail about why C. difficile is important and what methods are currently available for its detection to describe the need for the research presented here.

Response from authors: The abstract section will be elaborated to provide more background about current status of C.difficile diagnostic strategies. Thank you for this suggestion.

5. Introduction: In the previous GC-MS work on C. difficile, which compounds could be used to differentiate infected people from controls? Are they detected with SIFT-MS?

Response from authors : A recent study of stored stool samples from 53 cases and 53 controls, using thermal desorption-gas chromatography-time-of-flight gas chromatography, identified seven compounds (propan-1-ol, 3-methylbutanal, ethyl propionate, hexanoic acid, 4-methyl phenol, dodecane, and indole) indicative of CDI with a ROC >0.7. This information is provided in the introduction section. None of these molecules were part of the panel that was used in SIFT-MS. SIFT-MS is equipped to measure 22 common analytes that are present in breath in patients with disease states.

6. Table 1: Do the authors expect that the confounding variables from other conditions (e.g. cancer) will impact the differences between the groups? Data may need to be presented in the figures with color codes to indicate patients these other chronic diseases. 

Response from authors: Thank you for this query and suggestion ; Yes, we expect other disease states to affect VOC profile including cancer and metabolic conditions like diabetes mellitus. It is certainly possible that some of these comorbid conditions may be confounding factors. However, this was a pilot study and there were too few patients with each of these chronic conditions to evaluate if these chronic conditions were confounding factors. Future larger studies with larger sample size that is controlled for all confounding factors are needed. This point is added to the limitation section.

7. Why was the K nearest neighbors strategy selected? Please explain the rationale.

Response from authors: A KNN classifier was selected because it is a simple and intuitive classification tool. This being a pilot study with a small sample size we did not evaluate and compare different classification methods .

8. Do any of the molecules detected in SIFT-MS have potential as individual biomarkers? This point is addressed in part by the comparisons and p values in the supplemental tables, but it should also be presented in the results section of the manuscript.

Response from authors: Thank you for your suggestion. We have included Table 2 describing the concentration of VOCs in patients with and without CDI. None of the molecules on their own achieved statistical significance. Based on our findings, it does not appear that any of these compounds could serve as individual biomarkers that could effectively separate CDI from no CDI. but the pattern of VOC concentrations rather than individual concentrations was more meaningful. This is added to the discussion section of the manuscript.

9. Supplemental Figures and Tables need captions on the page with the data or image.

10. Supplemental Tables need headings to explain the data, which are presented as average and standard deviation but not labeled.

Response from authors: Supplementary Tables 1 and 2 are labelled with headings and data explained. Thank you for this suggestion.

Reviewer 3:

This paper would benefit if the authors would expand the introduction and addressed concerns below.

1.Introduction: This section is very short. It almost reads like an abstract. The authors should do a bett er job introducing breath analysis, its prior use for CDI, and the various analytical techniques. The section could end on the rationale used for the selection of SIFT-MS. This section should be at least two paragraphs longer.

Response from authors: Thank you for this suggestion. We have expanded the introduction section that now includes a problem statement why CDI is important, current diagnostic modalities and its limitations, VOCs in CDI and various methods that are utilized. We wanted to study usefulness of SIFT-MS as it was available in our center and no prior CDI studies have been done using this technology (mentioned in the introduction)

2.Materials and Methods:

a) Why were weekend samples excluded? Please do a better job explaining the rationale.

Response from authors: Research technicians worked only from Monday through Friday and hence weekend samples were not studied as it would not have been possible to have weekend samples analyzed promptly. 

b) Why were samples incubated before analysis? Is this common practice?

Response from authors: Thanks for this question. Samples were incubated to desorb the VOC's from the surface of the bag for analysis. Yes, this is a common practice in our VOC lab. This information is added to the methods section.

c) Table 1: Does not significantly contribute to the paper. I would recommend moving this to the supplemental section or shortening it for the main text. Response from authors: Thank you for this suggestion. Table 1 represents the baseline demographic data of cases and controls enrolled in this study. The authors feel that this is important to the study highlighting that these characteristics were not statistically different in 2 groups studied. But we will shorten it by removing lab features of cases and controls. Foot note is added to Table 1.

3.Supplemental Tables 1-3. Given that these numbers are at the heart of the study, I would move supplemental table 1 into the main text. For all supplemental tables, what are the units? What is pos and neg? CDI pos, CDI neg? Please label more clearly.

Response from authors: Thank you for this suggestion. We have moved supplemental table to the main text as table 2.

3. Supplemental Figure 1 and 2 need figure legends.

Response from authors: Thank you. We provided figure legends for supplementary figure 1 and 2

4. Results:

I understand that the entire 22 VOC panel was used for the ROC curves. However, was there a subset of VOCs in the panel that was MORE predictive than the full panel for each type of sample? If so what?

Response from authors: Thank you. No, we did not find a subset of VOCs that was more predictive than the full panel for each type of sample

Other general comments:

5.Why was blood plasma used? Why not whole blood? The sample processing of whole blood to plasma would significantly change and or remove important VOCs.

Response from authors: Thank you for this suggestion. Blood plasma was used as it was the most easily available left-over sample after daily testing. We agree that plasma extraction from the whole blood, might have affected the amount and number of VOCs and will include this as a limitation of the study.

---

## [Decision Letter · Decision Letter 1]

28 Apr 2021

PONE-D-20-23689R1

Diagnosis of Clostridioides difficile infection by analysis of volatile organic compounds in breath, plasma, and stool – a cross-sectional proof-of-concept study.

PLOS ONE

Dear Dr. John,

Thank you for submitting your manuscript to PLOS ONE. After careful consideration, we feel that it has merit but does not fully meet PLOS ONE’s publication criteria as it currently stands. Therefore, we invite you to submit a revised version of the manuscript that addresses the points raised during the review process.

Thank you for responding to the initial review.  There was still a major concern from one reviewer regarding the VOC analysis and in particular the biological aspects of measuring VOCs across the samples types.  It is critical that you carefully address this comment in you revision.  

We look forward to receiving your revised manuscript.

Kind regards,

Timothy J Garrett, PhD

Academic Editor

PLOS ONE

Reviewers' comments:

Reviewer's Responses to Questions

**Comments to the Author**

1. If the authors have adequately addressed your comments raised in a previous round of review and you feel that this manuscript is now acceptable for publication, you may indicate that here to bypass the “Comments to the Author” section, enter your conflict of interest statement in the “Confidential to Editor” section, and submit your "Accept" recommendation.

Reviewer #1: (No Response)

Reviewer #2: (No Response)

Reviewer #3: All comments have been addressed

2. Is the manuscript technically sound, and do the data support the conclusions?

Reviewer #1: No

Reviewer #2: Yes

Reviewer #3: Yes

3. Has the statistical analysis been performed appropriately and rigorously? 

Reviewer #1: I Don't Know

Reviewer #2: Yes

Reviewer #3: Yes

4. Have the authors made all data underlying the findings in their manuscript fully available?

Reviewer #1: Yes

Reviewer #2: (No Response)

Reviewer #3: Yes

5. Is the manuscript presented in an intelligible fashion and written in standard English?

Reviewer #1: Yes

Reviewer #2: Yes

Reviewer #3: Yes

6. Review Comments to the Author

Reviewer #1: Is there a reference on how the monitored VOCs were selected and their presence are common in breath, stool and blood? This is to give justification that the same VOCs monitored in breath can be monitored in stool and blood. Physically, the classes of samples differ in composition significantly, thus it is not obvious to the reviewer how the blood and stool samples are expected to contain the same VOCs found in breath.

The data presented does not support the conclusions claimed by the paper. It is egregious that the authors would attempt to monitor common breath metabolites (as stated in page 5 line 102) in stool and blood, and apply it as of same significance in presence and amount in contrast to breath samples. The previous question was not answered satisfactorily.

Reviewer #2: The authors have addressed most of the criticisms from the previous review, but a few minor changes need to be made.

The manuscript needs to be spellchecked to remove errors in the updated version.

For Figure 2, what sample set is used for the heat map? It would be helpful to center and normalize the data per analyte, so that you use more of the color spectrum. Now, everything looks yellow except for a couple of samples at the top, which are high outliers. Note that red/green designations will not be available to the color blind; red/blue is recommended or use of another indicator (bold font or -CDI appended after the sample name for positive cases would work well.

Reviewer #3: Thank you. All concerns have been addressed. If find this paper interesting and timely.

7. PLOS authors have the option to publish the peer review history of their article (what does this mean?). If published, this will include your full peer review and any attached files.

Reviewer #1: No

Reviewer #2: No

Reviewer #3: No

---

## [Author Response · Author response to Decision Letter 1]

16 Jul 2021

On behalf of the authors of the manuscript, ‘Diagnosis of Clostridioides difficile infection by analysis of volatile organic compounds in breath, plasma, and stool – a cross sectional proof-of-concept study’, we want to thank you and the reviewers for reviewing our manuscript and giving valuable suggestions. We have updated the manuscript to reflect the minor corrections suggested by the two reviewers.

Below are our responses to the specific points raised by the reviewers. 

Comment 1: Is there a reference on how the monitored VOCs were selected and their presence are common in breath, stool and blood? This is to give justification that the same VOCs monitored in breath can be monitored in stool and blood. Physically, the classes of samples differ in composition significantly, thus it is not obvious to the reviewer how the blood and stool samples are expected to contain the same VOCs found in breath Thank you for allowing us to present the revised manuscript. Author Response: The authors want to thank the reviewer for this great suggestion. We agree that we don’t have a solid justification on why we looked for the same 22 VOCs in three biologically different samples. This made us focus on the ‘full scan data’ set that includes all molecules or product ions that comes in the range of 14-200 mass to charge ratio values that are present in the sample compared to the ‘sim scan data’ which looks at only 22 VOCs. This analysis improved the ROC curves of all the 3 clinical samples we studied. Figures are updated in the re-submission

Comment 2: The manuscript needs to be spellchecked to remove errors in the updated version Author Response: Thank you for this suggestion. The manuscript has been spell checked. 

Comment 3: For Figure 2, what sample set is used for the heat map? It would be helpful to center and normalize the data per analyte, so that you use more of the color spectrum. Now, everything looks yellow except for a couple of samples at the top, which are high outliers. Note that red/green designations will not be available to the color blind; red/blue is recommended or use of another indicator (bold font or -CDI appended after the sample name for positive cases would work well.

Author Response: Thank you for this suggestion. We used breath product ion concentration to build the heatmap. As per the suggestion from the reviewer, we changed the color spectrum to red and blue color. 

Thank you for allowing us to review this manuscript. Hope you find the revisions acceptable.

---

## [Editor Report · Decision Letter 2]

4 Aug 2021

Diagnosis of Clostridioides difficile infection by analysis of volatile organic compounds in breath, plasma, and stool – a cross-sectional proof-of-concept study.

PONE-D-20-23689R2

Dear Dr. John,

We’re pleased to inform you that your manuscript has been judged scientifically suitable for publication and will be formally accepted for publication once it meets all outstanding technical requirements.

Kind regards,

Timothy J Garrett, PhD

Academic Editor

PLOS ONE

Additional Editor Comments (optional):

I appreciate your clear and scientific valuable additions made after the second revision. The key aspect was providing clarity on VOCs in plasma, stool and breath and I think you have clearly done that. At this point, you have addressed all major concerns regarding the publication of this work.
---

## [Editor Report · Acceptance letter]

9 Aug 2021

PONE-D-20-23689R2 

Diagnosis of *Clostridioides difficile* infection by analysis of volatile organic compounds in breath, plasma, and stool – a cross-sectional proof-of-concept study. 

Dear Dr. John:

I'm pleased to inform you that your manuscript has been deemed suitable for publication in PLOS ONE. Congratulations! Your manuscript is now with our production department. 

Kind regards, 

on behalf of

Dr. Timothy J Garrett 

Academic Editor

PLOS ONE